# Antibiotic Resistance and Food Safety: Perspectives on New Technologies and Molecules for Microbial Control in the Food Industry

**DOI:** 10.3390/antibiotics12030550

**Published:** 2023-03-10

**Authors:** Jannette Wen Fang Wu-Wu, Carolina Guadamuz-Mayorga, Douglas Oviedo-Cerdas, William J. Zamora

**Affiliations:** 1Food Technology Department, University of Costa Rica, San José 11501-2060, Costa Rica; 2Agribusiness Department, Costa Rican Technological Institute, Cartago 159-7050, Costa Rica; 3Instrumental Analysis Department, Iberoamerican University, Tibás 11870-1000, Costa Rica; 4CBio^3^ Laboratory, School of Chemistry, University of Costa Rica, San Pedro, San José 11501-2060, Costa Rica; 5Laboratory of Computational Toxicology, University of Costa Rica, San Pedro, San José 11501-2060, Costa Rica; 6Biological Testing Laboratory (LEBi), University of Costa Rica, San Pedro, San José 11501-2060, Costa Rica; 7Advanced Computing Lab (CNCA), National High Technology Center (CeNAT), Pavas, San José 1174-1200, Costa Rica

**Keywords:** antibiotic resistance, food chain, antimicrobial peptides, food safety, food pathogens

## Abstract

Antibiotic resistance (ABR) has direct and indirect repercussions on public health and threatens to decrease the therapeutic effect of antibiotic treatments and lead to more infection-related deaths. There are several mechanisms by which ABR can be transferred from one microorganism to another. The risk of transfer is often related to environmental factors. The food supply chain offers conditions where ABR gene transfer can occur by multiple pathways, which generates concerns regarding food safety. This work reviews mechanisms involved in ABR gene transfer, potential transmission routes in the food supply chain, the prevalence of antibiotic residues in food and ABR organisms in processing lines and final products, and implications for public health. Finally, the paper will elaborate on the application of antimicrobial peptides as new alternatives to antibiotics that might countermeasure ABR and is compatible with current food trends.

## 1. Introduction

The introduction of antibiotics as a means to treat and control infectious diseases marked a turning point in the evolution of modern medicine [1]. However, misuse of these antimicrobial agents has caused the development of widespread antibiotic resistance (ABR) in organisms ranging from spoilage microorganisms to pathogens [2,3].

ABR is a growing concern not only in the medical and veterinary fields but also in the food industry, as it compromises the quality and safety of the food supply chain. ABR is one of the biggest threats to global health, food security, and development [4]. The causes of ABR are complex and include inappropriate prescribing practices, patient behavior, inadequate patient education, the unauthorized sale of antimicrobials, the lack of drug regulatory mechanisms, and excessive use of antimicrobials in animal production [5]. The misuse of antibiotics is often more significant in less developed countries where the lack of education can lead to the incorrect application of drugs, dosing, and waiting times [5,6] Accordingly, the consequences of ABR are often felt more severely in less developed countries where healthcare systems are more vulnerable [1,3].

A promising alternative to conventional antibiotic treatments is the use of antimicrobial peptides (AMPs). AMPs are short peptides (15–20 amino acids) that are highly cationic and hydrophobic. They have broad spectrum fast action, a rapid killing rate, and generic membrane and intracellular effects that do not target specific molecules [7,8,9] Their mechanisms of action, which are complex and harder to counter than those of antibiotic drugs, make the development of resistance difficult [9].

The present review focuses on ABR in the food supply chain, its legal implications, and future alternatives based on AMPs to mitigate the risks associated with the use of antibiotics in the food production chain.

## 2. Methodology

This paper was developed through qualitative examination of publications between 2001 and 2022 on antibiotic resistance and food safety. The keywords “antibiotic resistance”, “food supply chain”, “mechanism of resistance”, “antimicrobial peptides”, “alternatives to antibiotics”, “AMPs”, and “food safety” were searched on Google Scholar, PubMed, Web of Science, and the internal Data Base of the University of Costa Rica. The information was selected and extracted following these criteria: (a) processing (e.g., processing line, unit operations, workers), (b) type of product (animal source, plant source, processed food, raw food), and (c) timeline. All available information was compiled and tabulated for qualitative analysis. The data was assessed by multiple individuals and compiled and reviewed by the first and corresponding author.

## 3. Antibiotic Resistance in the Food Chain

Antibiotics are natural, synthetic, or semi-synthetic substances with the capability to inhibit the growth of microorganisms associated with human and animal diseases [1,10]. Antibiotic resistance is a natural phenomenon that occurs when antibiotics that were initially effective against certain bacterial infections can no longer inhibit the growth and development of the causal microorganisms [3,11,12]. When microorganisms are exposed to antimicrobial drugs, susceptible bacteria are killed or inhibited, while bacteria that are intrinsically resistant or have acquired a resistance trait have greater chances of survival and proliferation due to selective pressure [13]. Antibiotic resistance is promoted mainly through the overuse of antibiotics; however, other practices such as inappropriate use, inadequate dosing, and poor adherence to treatment guidelines also contribute to the growth of resistance phenotypes [1,4,13].

Globally, the increasing prevalence of antibiotic resistance in a broad range of microorganisms threatens human and animal health [3,11,12] A growing number of treatments have failed in patients with infections caused by multi, extensive, and pan-drug resistant bacteria [4,14]. When antibiotics generally used as first-line treatment are no longer effective, it becomes necessary to use last-resort options that are often more expensive and/or toxic. As a result, treatment of the disease becomes complex. The burden caused by antibiotic resistance is felt more strongly in low- and middle-income countries where healthcare systems have fewer economic resources and often lack the tools for proper disease diagnosis [15].

### 3.1. Acquisition of Antibiotic Resistance and Mechanisms of Transfer

Resistance genes can be found within the bacterial core genome (intrinsic resistance), or they can be acquired from an external source through lateral gene transfer (extrinsic). These genes can be exchanged when bacteria interact throughout the different steps of the food supply chain.

There are many ways by which bacteria acquire extrinsic resistance to antimicrobial agents through genetic exchange. The two main pathways of gene transfer in bacteria are (a) vertical transfer, which is the transfer of genes from parent cells to daughter cells through replication, and (b) horizontal transfer, which is the transfer of genes between different subjects through mobile elements [16].

The acquisition of genes through horizontal gene transfer requires three conditions. First, a vector is needed to deliver the donor DNA to the recipient cell. For example, a preexisting gene sequence can be modified by mutation to code for a resistance mechanism [17]; the modified gene can, then, be acquired through gene exchange. Bacteria are known to be “genetically promiscuous”, and gene flow is a recurrent process between organisms regardless of genus or species [17]. Second, the foreign DNA must be assimilated in the genome of the recipient or become associated with an autonomous replicating element like a plasmid. Finally, the foreign DNA must be expressed in a way that benefits the recipient cell [17].

There are three mechanisms of horizontal gene transfer: transformation, transduction, and conjugation. Transformation involves the import of exogenous free DNA from the environment and its incorporation into the main genome through homologous recombination [17,18]. This mechanism can potentially transfer DNA between very distantly related organisms. It is facilitated by competence machinery encoded by the bacteria, therefore, extent of genes acquirable through this method will depend on the bacteria. Some bacteria are continually competent to accept foreign DNA, while others are only competent at one point in the life cycle. Transformation is limited by the specificity of the DNA absorbed and the size of the chain.

Transduction is the introduction of foreign DNA by a viral vector (bacteriophage) that infects the bacterial cell, replicates during the lysogenic phase, and DNA fragments randomly into the chromosome (generalized transduction) or in a location adjacent to the phage attachment site (specialized transduction) [17,19]. This mechanism is highly specific, as it depends on receptors recognized by the bacteriophage. The size of the DNA transferred in a single event is limited to the capsid size of the phage [19].

Conjugation involves physical contact between donor and recipient cells and a self-transmissible or mobilizable plasmid [17,19]. Of the three mechanisms for horizontal gene transfer, only conjugation requires direct interaction between the cells. This mechanism allows the transmission of large-sized DNA material and is considered the main mechanism of transmission of ABR.

Despite the diversity of mechanisms for gene transfer, the process is not successful unless the transferred sequences are assimilated and maintained in the recipient microorganism. This is usually achieved by mobile elements. Mobile elements are genomic sequences, such as plasmids, prophages, pathogenicity islands, restriction and modification systems, transposons, and insertion sequences, that can be transmitted vertically during cell division or by horizontal transfer [20]. Transmission of ABR genes is commonly mediated by plasmids since these structures are rarely integrated into the chromosome. Plasmids are small circular DNA molecules found in bacteria and some other microorganisms. They are physically separate from chromosomal DNA and can replicate independently so that several copies may be present at the same time [21]. The traits carried on a plasmid must confer an advantage of sufficient significance to avoid elimination from the cell [3,17,22].

ABR genes can also be transmitted by transposable elements or propagated by integrons [23]. Transposable elements are mobile sequences that can insert into several sites in the genome and cause deletions or mutations [24] Integrons are genetic elements located in gene cassettes that can be rearrange in open reading frames, allowing their expression [25]. Integrons can mediate the evolution of bacteria by acquiring, storing, disposing, and resorting the reading frames of each cassette; and some may contain a collection of genetic cassettes that encode for antibiotic resistance [26]. Resistance integrons have several common features, including motion and length, and only carry antibiotic-resistance genes in most of the cases [26].

Due to the nature of the food supply chain, microorganisms can easily interact at different points in the process. This favors the exchange of genetic material and the acquisition of resistance genes.

### 3.2. Mechanisms of Antibiotic Resistance

Before addressing some of the main issues regarding ABR in the food industry, it is important to define the concepts of persistence and resistance. In both cases, a small population of individuals withstands the antimicrobial treatment. When a bacterium is resistant, its daughter cells inherit the resistance. On the other hand, persistence describes bacterial cells that are not susceptible to the antimicrobial but do not possess resistance genes that are transferable to their daughter cells [19]. The survival of persistent cells occurs mainly because some cells in the population are in the stationary growth phase, and most antimicrobials have little to no effect on cells that are not actively growing and dividing [22]. In contrast to ABR cells, persistent cells become susceptible to the antibiotic when they enter the growth stage to establish a new population.

Four main mechanisms of ABR in bacteria are (1) Limiting drug uptake; (2) modification of the drug target; (3) inactivation of the drug and (4) active drug efflux [19] as shown in Figure 1. These mechanisms apply to either acquired or natural resistance and may vary depending on the cell structure. Drug uptake limitation, drug inactivation, and drug efflux are common natural resistance mechanisms, while drug target modification, drug inactivation, and drug efflux can be easily acquired [23].

#### 3.2.1. Drug Uptake Limitation

The mechanism of drug uptake limitation is often involved in natural resistance. This mechanism depends mainly on the structure and composition of the bacterial cells. Due to chemical affinity or size, the cell may be less permeable to antimicrobial molecules and therefore less susceptible to their effect [19,23].

#### 3.2.2. Drug Target Modification

The modification of antibiotic targets is a resistance strategy prevalent in many organisms by which the structure targeted by the antibiotic is altered enzymatically. Since antimicrobials are designed to inhibit growth through the degradation or inactivation of specific cell structures, when those structures are modified and are not recognized by the agent and the antimicrobial will no longer have the original effect [23,28]. An example of this is resistance to glycopeptide. This antibiotic acts at the outer leaflet of the cell membrane by binding to Lipid II and blocking the synthesis of peptidoglycan. Resistance results from the modification of a pentapeptide stem in Lipid II that leads to the loss of the hydrogen bond donor and the introduction of electrostatic repulsion between glycopeptide and the peptide stem, which lowers the affinity of the drug to the cell [28]. Another example is polymyxin resistance, which involves the modification of lipopolysaccharide (LPS) in the membrane of gram-negative bacteria [28].

#### 3.2.3. Drug Inactivation

There are two main paths by which bacteria inactivate drugs. One is the direct degradation of the drug and the second is the transfer of a chemical group to the drug that alters its structure and hence its functionality. The degradation of drugs is mainly mediated by enzymes like the β-lactamases, which are responsible for resistance to β-lactam drugs [29]. These enzymes prevent interaction between the target and the drug by modifying the drug’s binding sites.

The inactivation of drugs by chemical group transfer commonly uses acetyl, phosphoryl, and adenyl groups through transferases. One of the most common modifications is acetylation which is known to be used against aminoglycosides, chloramphenicol, streptogramins, and fluoroquinolones. Other mechanisms such as phosphorylation and adenylation are primarily used against aminoglycosides [23,28,29].

#### 3.2.4. Drug Efflux

Bacteria possess chromosomally encoded genes for efflux pumps. Efflux pumps are cytoplasmic membrane protein complexes that transport harmful substrates such as dyes, chemicals, and antibiotics out of the cells [28]. Efflux pumps are classified into five main families based on their structure and energy source: (1) ATP-binding cassette (ABC) family; (2) multidrug and toxic compound extrusion (MATE) family; (3) small multidrug resistance (SMR) family; (4) major facilitator superfamily (MFS), and (6) the resistance-nodulation-cell division (RND) family [28].

##### ABC Transporter Family

The ABC efflux family includes both uptake and efflux transport systems. This family transports amino acids, drugs, ions, polysaccharides, proteins, and sugars using ATP as the energy source. These pumps have specific substrates and have been linked to resistance to fluoroquinolones and tetracyclines [28,29].

##### MATE Transporter Family

The MATE efflux family uses gradients of Na^+^ as an energy source. The primary function of this family is to move cationic dyes and fluoroquinolone drugs. They can also efflux aminoglycosides and other unrelated chemical structures [30].

##### SMR Transporter Family

The SMR efflux family uses proton-motive force (H^+^) as energy. They are hydrophobic and efflux primarily lipophilic cations. The genes for these pumps have been found in both chromosomal and movable DNA such as plasmids and transposable elements. This family has been linked to resistance to β-lactams and some aminoglycosides [28].

##### MFS Transporter Family

The MFS efflux family catalyzes transport via solute/cation (H^+^ or Na^+^) symport or solute/H^+^ antiport and has been linked to the transport of anions, drugs such as macrolides, and tetracyclines, metabolites, and sugars. This family has the greatest diversity in the substrates; however, individually they are more specific. They have been linked to resistance to erythromycin, chloramphenicol, macrolides, fluoroquinolones, and trimethoprim [30,31].

##### RND Transport Family

The RND efflux family catalyzes substrate efflux through a substrate/H^+^ antiport mechanism that is widely distributed in gram-negative bacteria. They are involved in the efflux of antibiotics, detergents, dyes, heavy metals, solvents, and other substrates. Some can be drug or drug-class specific, but many of these pumps can transport a wide range of drugs and components of similar chemical structures [30,31,32].

## 4. Potential Routes of Transmission and Prevalence of ABR in the Food Chain

Consumers can potentially be exposed to ABR bacteria at many points within the food chain. It is also possible for bacteria to interact and exchange genetic material throughout the food supply chain. According to EFSA [33], the extent of exposure to antimicrobial-resistant bacteria through the food chain is difficult to determine and the role of food in ABR gene transfer has been insufficiently studied. However, the occurrence of ABR organisms in food would have an impact on humans. There are many opportunities for transmission between animals, food handlers, and the environment throughout the food chain (Figure 2).

Antimicrobials have become indispensable for decreasing morbidity and mortality associated with infectious diseases. Animal health and productivity have improved significantly over the past several decades due to the introduction of antimicrobials into veterinary medicine [34]. Despite emerging resistance to these molecules, antibiotics are still effective for the control of most infectious diseases; however, the loss of efficacy due to bacterial antimicrobial resistance is becoming more common [34], and transmission of ABR microorganisms to humans as foodborne contaminants is a threat to public health. Resistance mechanisms have been identified and reported for all antimicrobials currently available for clinical use in both human and veterinary medicine (Table 1).

In food animals, antibiotics are predominantly used in intensive livestock farming to treat respiratory and enteric infections. They are also administered at sub-therapeutic levels in concentrated animal feed to promote growth, improve feed conversion efficiency, and prevent disease [35].

In 1997, the World Health Organization (WHO) declared that the overuse of antimicrobials could lead to the selection of resistant forms of bacteria in the ecosystem and recommended that antibiotics essential to human treatments should not be used as growth promoters in animals [10]. The constant addition of antibiotics to feed or water to control bacterial infections in intensive fish farming is another source of exposure of wild microorganisms to antibiotics [36,37]. Most antibiotics administered to livestock are not fully metabolized and are released along with their transformation products into the environment through feces and urine [37]. This organic waste can reach soil through natural means or in compost and subsequently contaminate soil, crops, and water sources. Exposure of susceptible bacteria to these antibiotics could lead to the development of resistance [37].

Antibiotic residues in food have been associated with many health issues. Several antibiotics are reported to cause skin allergies and anaphylaxis (e.g., penicillin) [38] mutagenesis and blood dyscrasia have been associated with the use of sulfonamides and chloramphenicol [39,40]. Chloramphenicol has also been associated with anomalies in bone marrow activity [40]. The Food and Drug Administration (FDA) closely regulates the use of drugs that can cause hypersensitivity or are toxic or carcinogenic, including antibiotics such as chloramphenicol, sulfonamides, and fluoroquinolones [41]. Antibiotic residues in foods can also affect the gut microbiota and can lead to dysbiosis that may cause other related diseases [42,43]. Most of these effects are more significant when the exposure to antibiotics is prolonged and continuous, as in the case of antibiotic residues. The most immediate risks are related to allergies and modifications of the gut microbiota, which may cause acute symptoms [40,43]. As seen in Table 2, antibiotic residues have been reported in various food products [44,45,46,47,48,49]. The presence of these residues is likely to induce and accelerate the development of antibiotic resistance and promote the transfer of ABR genes, while long-term exposure may lead to other pathologies.

Antibiotic residues are predominantly found in animal products, but crops can also be contaminated through irrigation and soil. The most common food crops that accumulate antibiotics are cereals, such as wheat, rice, and oats, and coarse grains, such as maize and barley [50]. Foods of animal or plant origin carry their own microbiome and thus constitute a potential route for the transmission of resistant bacteria and ABR genes between food, animals, and people due to their interactions. Potential recontamination of food products is possible at various stages in the food chain, threatening the product’s safety. This contamination can happen with more or less “harmless” organisms that cause spoilage or much more dangerous organisms such as foodborne pathogens such as Salmonella and Campylobacter.

Using data on the prevalence of ABR in isolates from human and food samples, a recent study determined through meta-analysis that the mean prevalence of ABR foodborne pathogens isolated from food was ≥11%, and most of the foodborne pathogens showed high resistance to β-lactams [51]. Multi-drug resistant pathogens were prevalent in >36% of all food types studied with the highest rates in meat products. Resistance to β-lactams was the most prevalent. Aquatic products showed a higher prevalence of fluoroquinolone and sulfonamide resistance. Pathogens with ABR genes isolated in other studies from animal-derived food products and humans are shown in Table 3. This analysis considered only pathogenic organisms and did not include other microorganisms that might show resistance and potentially transfer resistance genes. This is an important issue though, because due to chemical similarities between the molecules, antibiotic resistance may be associated with resistance and/or tolerance to disinfectants [52]. The presence of antibiotic residues in foods and resistance genes in pathogenic and spoilage bacteria increases the potential for gene transfer in the food chain.

## 5. Antibiotic Resistance and Food Safety: Implications for Public Health

Food safety is a critical aspect of the food supply chain as it inherently defines the quality of the final product and hence, consumer acceptance. Adoption of the Hazard Analysis and Critical Control Points (HACCP) system brought about a monumental change to the food industry. This new by-design approach to food safety considers all aspects of food processing in an integrated controlled safety assurance system for hazard prevention and control [57] known as the “Food Safety Management Program” whose main aim is to ensure consumer safety [58]. Many industries currently follow the program for food supply control. These programs consider all risks associated with processes that could threaten the safety of the final product. One of the most common risks is accidental or unintended microbial or chemical contamination of the product within the production chain. However, these programs usually do not address the possibility of adulteration or voluntary contamination [59], which can be more challenging to detect.

Food fraud, also referred to as economically motivated adulteration (EMA), is defined as any kind of intervention such as the adulteration, deliberate and intentional substitution, dilution, simulation, alteration, falsification, or mischaracterization of a product, its ingredients, or packaging, or the use of false or misleading information about a product to obtain an economic gain. These practices have ancient origins and are common in less regulated markets; however, these illegal practices have expanded due to factors such as globalization and market internationalization. The complexity of the food industry makes verification of product integrity difficult to follow during its traceability. Products can be easily adulterated during transit from the manufacturer to the final consumer. Food fraud can directly threaten food safety and economics, as seen in high-profile cases like the addition of melamine to dairy products in China (2008) or the presence of fipronil in eggs (2017). It can also be an indirect threat to consumers continuously exposed to unauthorized ingredients like antibiotics. Antibiotic residues in food have become a significant hazard and an example of food fraud. Antibiotic residues are present in several products of daily consumption, mostly from animal sources, as noted in Table 2, but they have also been detected in processed products sold openly on the market. For example, in 2009, large shipments of honey with erroneous label information were imported into the United States; despite testing positive for antibiotics, the product continued to be sold [60].

Antibiotic residues in food are more common in developing countries where education and regulations tend to be less rigorous than in developed countries. Uniform regulation of antibiotic use in agriculture is challenging since practices vary significantly between regions. Nonetheless, international, and local regulatory authorities, like the World Health Organization (WHO) and the European Food Safety Authority (EFSA), have made efforts to establish standards based on each country’s context for better regulation of antibiotic use [61]. The harmonization of those standards is based on parameters such as (1) acceptable daily intake (ADI), which is a toxicological standard; (2) withdrawal period or “Waiting Time” (WT), which refers to the minimum time from the administration of the last dose to the production of the food, and (3) Maximum Residue Level (MRL) [62]. Although the ADI, WT, and MRL have been established for many antibiotics, and there is a significant effort to control MRL worldwide through regulations promoted by the World Trade Organization and the Codex Alimentarius, control of antibiotic residues remains difficult since MRL values are mostly geographically dependent [61,62]. The lack of policies regarding the use and misuse of antibiotics in both humans and animals has allowed ABR to become a serious threat [11].

ABR is a problem of high importance not only in the food industry but also for public health. Since 2015, AMR has become a worldwide priority. The WHO created the “Global Action Plan on Antimicrobial Resistance” to encourage the wise use of antibiotics and propose strategies to reduce their consumption. This global plan identified several concerning health risks. The greatest concern was that some common medical conditions associated with bacterial infections, such as tuberculosis, sexually transmitted diseases, urinary tract infections, pneumonia, bloodstream infections, and foodborne diseases have become more difficult to treat since the causal organisms show resistance to a large number of conventional antibiotics [63]. As more resistance mechanisms evolve, microorganisms acquire the ability to survive current antibiotics. Another consequence of ABR is multi-drug resistance (MDR). MDR bacteria are resistant to multiple antibiotics simultaneously. MDR organisms are a high threat to public health, and the number of MDR organisms continues to increase. Among the most common microorganisms are *A. baumannii*, *E. coli*, *P. aeruginosa*, *K. pneumonia*, *S. aureus*, *S. pneumonia*, *E. faecium* and *E. faecalis*. MDR organisms can lead to the deterioration of healthcare systems; diseases caused by MDR bacteria may be more severe and have increased mortality rates, and the irresponsible use of antibiotics in empirical or specific treatments could have effects at both the pharmacological and economic levels of the public health system [43].

At present, most countries lack systems for surveillance of antibiotic use. A regulatory framework is necessary at both local and international levels to assess the risks and benefits of using antibiotics. This framework must be comprehensive and supported by standards, guidelines, and recommendations for the effective control of antibiotic use in the food chain [2]. Despite advances in analytical methods for the detection, identification, and isolation of food-borne microbes, food safety is still traditionally based on a finished product-testing approach primarily for detecting possible hazards at the end of the processing line. Finished-product sampling is valuable in situations such as traditional lot testing for withhold/release verification [64]. There are established criteria in standards and other legal frameworks for microbiological and chemical hazards. However, most of these focus on product safety and revolve around processing, often leaving aside concerns that are not inherently microbiological, for example, environmental aspects and contamination with antibiotics [64]. The most common method for detection of antibiotics in food is chemical analysis of the final product. These analyses aim to identify and quantify the antibiotic residues and compare it to a reference criterion to determine its compliance and can be used as routine controls for quality assurance in the industry. The most frequently used techniques include High-Performance Liquid Chromatography (HPLC) and Mass Spectrometry (MS). The high sensitivity of these methods allows the quantification of trace levels (nanograms per gram) of antibiotic residues in samples. These methods are very accurate, but usually only the final product is analyzed, and the presence of antibiotic residues is not monitored at each of the processing steps. Considering the emerging issue of ABR and the potential changes to these molecules that may occur during processing, the presence of these contaminants should be monitored throughout processing as a control in the food industry [65].

Antibiotic contamination can be associated with severe adverse consequences involving four primary levels: (1) animal health, (2) environmental; (3) transformation process, and (3) consumer health [61]. Antibiotics can accumulate in edible crops, drinking water, and animal products in the form of both antibiotic compounds, or degradation products. In a study performed in China, researchers identified 58 antibiotics in drinking water and 49 in food samples, estimating a probable daily intake of about 310,200 and 130 ng/kg-bodyweight in children, teenagers, and adults, with a maximum of 1400. 970 and 530 ng/kg-bw/day [66]. Their presence in food can cause mild to adverse complications that can be divided into (1) direct toxicity and allergic reactions, and (2) resistance to antibiotics [67]. Antibiotic residues can act as allergens that elicit allergic reactions with symptoms such as skin rashes, serum sickness, thrombocytopenia, erythema multiforme, hemolytic anemia, vasculitis, acute interstitial nephritis, Stevens–Johnson syndrome and toxic epidermal necrolysis [42]. Allergic reactions associated with antibiotic residues have been reported in people who consumed contaminated milk [42] and meat [38,39]. The presence of antibiotic residues in food had also been potentially linked to hepatotoxicity [68,69], carcinogenesis, mutagenesis, reproductive disorders, and teratogenicity [67]. Additionally, the presence of antibiotic residues in food and animal feed may affect the gut microbiome causing dysbiosis that can lead to problems such as obesity, intestinal barrier damage, and increased food allergies [70].

All these implications punctuate the importance of developing new molecules to combat microbial infections as an alternative treatment to antibiotics, preventing the apparition of resistance to synthetic, semi-synthetic, or natural antibiotics. Some examples of these include the development of nano-delivery systems such as liposome nanoparticles of gold, silver, zinc, and copper, which combined with drugs can create a synergistic antibacterial [28], or the use of therapies with targeted drugs such as bacteriocins. The application of new techniques based on targeted therapies using peptides has fewer side effects in terms of toxicity compared to metal compounds in the liposome, which gives them a superior advantage as a new alternative to combat antibiotic resistance.

## 6. New Alternatives to Antibiotics: Bacteriocins and Their Physicochemical Properties

Antibiotic resistance has made it urgent to search for alternatives with novel modes of action that are less likely to lead to bacterial resistance. Intensive study in the biopharmaceutical industry is focused on a novel class of compounds with potential therapeutic properties. These molecules are known as antimicrobial peptides (AMPs). Antimicrobial peptides are bioactive, small proteins that are produced naturally by some living organisms as indispensable components of their innate immune system, becoming the first-line defense against microbial attacks in Eukaryotes; or produced as a competitive strategy in Prokaryotes to limit the growth of other microorganisms [71]. They are also known as host defense peptides and can be classified depending on electronegativity, structure, or synthesis pathways (ribosomal or non-ribosomal). These peptides are produced in lower and higher organisms, and their synthesis is cell-specific and may be constitutive or inducible in response to “challenge” stimuli, and commercially, they can also be synthesized through bioengineered or chemical ways [72,73]. AMPs’ primary role is killing invading pathogens.

AMPs have antimicrobial properties, which have allowed their use as natural alternatives to chemical additives for shelf life and food safety and are, nowadays, used extensively in several products. At the same time, these small protein molecules have also shown promising properties in the treatment of infectious diseases. Conventional antibiotics often target bacteria based on their antibacterial activity, which can eventually lead to ABR; meanwhile AMPs interact with bacterial cell membranes through different means, leading to its dead [71,74] Several proposed mechanisms explain the permeabilization of bacterial membranes by AMPs. However, generally speaking, the effect has been primarily attributed to their positive charge that allows these peptides to interact with components of the bacterial cell, resulting in the disruption of the lipidic bilayer, leading to cellular death [75]. There are other non-membranolytic mechanisms based mainly on intracellular activities such as nucleic acids, proteins, or cell wall synthesis inhibition [76]. Since AMPs can be produced for a variety of organism, it is important to clarify that this section will only address AMPs produced by bacteria.

### 6.1. Bacteriocins

Bacteriocins are a specific kind of ribosomally-synthesized AMPs of a length of 20–60 amino acids, cationic and hydrophobic, produced by many species of bacteria and archaea [77,78]. These peptides have shown promising potential in the food industry (see Table 4). Studies have noted that antimicrobial peptides can act as bioprotectors against spoilage and pathogen contamination since they have shown excellent antimicrobial activity against gram-positive and gram-negative bacteria. Additionally, they prevent the proliferation of thermophilic, spore-forming microorganisms [72,79]. Nowadays, one of the most relevant safety problems in the food industry is cross-contamination with bacteria such as *Salmonella* spp., *Shigella* spp., *Micrococcus* spp., *Enterococcus faecalis*, *Bacillus licheniformis*, *Escherichia coli*, *Listeria monocytogenes*, *Staphylococcus aureus*, *Campylobacter jejuni*, *Yersinia enterocolitica*, *Vibrio parahemolyticus*, *Escherichia coli* 0157:H7, and *Clostridium botulinum* [79], and due to concerns regarding synthetic additives usage and consumers’ growing interest in clean-label products, the use of alternative natural ingredients has gained a pivotal role. For instance, the use of lactic acid, a safe agent for food preservation approved by the United States Food and Drug Administration (USFDA), as well as hydrogen peroxide, and some peptides produced in the fermentation process are commonly used bio-preservatives. For example, Nisin, a bacteriocin produced by *Lactococcus lactis,* is a legally approved natural preservative for dairy products, canned vegetables, juice, alcoholic beverages, meat, and fish used to prevent food-spoilage caused by *Lactobacillus* spp, and prevents the growth of *L. monocytogenes, S. aureus* and *Clostridium* spp. [77], also increases shelf-life without changing the flavor, texture or aroma, particularly does not alter the physical, chemical and biological properties [79]. Nisin has also been approved for clinical use as an alternative to antibiotics due to its broad spectrum against both gram-negative and gram-positive pathogens [80] and several studies have reported its effectiveness for treating infections such as mastitis [81], respiratory diseases [82] and skin infections [83], which makes it a potential substitute for veterinary use.

### 6.2. Bacteriocin Classification

Bacteriocins classification can be performed based on various properties, considering amino acid composition, the type of post-translational modifications, and peptide size. These diverse criteria make bacteriocin categorization very wide and variable. Nonetheless, due to the recent interest in their potential applications, the classification system is in a constant improvement [88]. Bacteriocins are produced by both gram-negative, gram-positive, and a few archaea, and each group contains its own classification structure.

#### 6.2.1. Classification of Gram-Negative Bacteriocins

Gram-negative bacteriocins are grouped into two categories: colicins and microcins. The genes encoding for these peptides can be contained in a plasmid or the chromosome in a set of three genes that includes a structural gene, an immunity protein gene, and a lysis gene involved in the peptide’s final secretion [88]. At the same time, colicins can be classified into two classes, based on translocation, and into three classes, if based on their action modes. Microcins are grouped into two classes: I y II; the latter is subdivided into two subclasses: a and b [89].

#### 6.2.2. Classification of Gram-Positive Bacteriocins

Gram-positive bacteriocins are classified into three classes based on their biochemical and genetic properties. At the same time, Classes I and II have two and three subclasses, respectively. However, these bacteriocins lack a systematic organization, thus why a modern classification based on structural similarity, phylogenetic evolution, and consensus motif sequence was proposed [90]. Using this modern classification, gram-positive bacteriocins are grouped into 12 groups, where 1, 3, and 4 are further divided into subgroups a and b [88,90].

#### 6.2.3. Classification of Archaea Bacteriocins

Several archaea have been reported to produce bacteriocins, and, to date, two types of archaeocins, halocins, and sulfolobicins have been identified and described [91,92]. Halocins are produced by all members of the halobacteria group, providing a great diversity within these molecules. Based on their size, they can be further classified into microhalocins (3.4 kDa) and larger ones (35 kDa). Sulfolobicins are produced by Sulfolobus islandicus, and this bacteriocin has a very narrow spectrum, inhibiting only other members of the genre [91].

### 6.3. Bacteriocin Synthesis

The genes for their synthesis are usually contained in operon clusters, harbored in the genome, or contained in plasmids or mobile elements [78]. The expression of these genes is inducible, and it is often led by the presence of an auto-inducer peptide (e.g., the bacteriocin itself) [93]. The expression is usually regulated by a two-component or three-component regulatory system; nonetheless, some bacteriocins can show unique ways of the regulation of its expression [94]. Bacteriocins are synthesized (Figure 3) as precursors, which are later post-translationally modified. These modifications and eventual secretion, are led by accessory genes that are proximal to the gene that encodes for the bacteriocin precursors [95]. Subsequently, modified bacteriocins are transported and cleaved to generate the mature form [94,95]. The mature bacteriocin is then secreted by transporters such as the ABC transporters and sec-dependent exporters to be released extracellularly [78].

Precursor bacteriocins are biological inactives consisting of an N-terminal leader peptide attached to the C-terminal peptide. The leader peptide serves as the recognition site that directs the propeptide towards maturation and transport proteins; it also protects the producer strain by keeping the bacteriocin in an inactive form within the cell and interacts with the propeptide domain to ensure its conformation is suitable for enzyme-substrate interaction [96].

Bacteriocins produced through microbial fermentation means and their subsequent downstream processing often lower the final product’s yield. This limitation creates a challenging scenario for the molecule’s industrialization.

The alternate approaches were developed using solid-phase peptide synthesis (SPPS) and synthetic biology. This mechanism allows the reduction of production costs, and the advances in chemical biology have allowed the chemical process to mimic natural conditions [88]. Chemical synthesis provides the possibility to use unnatural amino acids, introduce pseudo-peptide bonds and perform side-chain modifications, giving them a wider range of properties compared to the ones obtained through genetic engineering methods [97].

#### Solid-Phase Chemical Synthesis (SPPS)

In the SPPS method, the N α-protected amino acids are attached to the N-terminal amine of the growing peptide chain, bound on the solid support. This is followed by the deprotection of the amino group to continue with the elongation of the chain in a two-step cycle of coupling and deprotection. This mechanic is repeated until the peptide sequence is completed, then the desired peptide is released from the solid support, and the sidechain protecting groups are removed to obtain the peptide of interest [97,98].

There are two strategies for SPPS: linear (Figure 4a) and convergent (Figure 4b) synthesis. The linear or sequential synthesis approach involves the stepwise addition of amino acids until de peptide of interest is produced; this process is limited to peptides with 50 residues in length [97].

Convergent synthesis, on the other hand, involves the independent SPPS of peptide fragments that are later cleaved from the polymer and linked through condensation reactions on a solid or in a solution with a standard coupling reagent of chemoselective reactions. Contrary to linear synthesis, convergent strategies allow the synthesis of peptides with longer chains (>50 residues) [99].

### 6.4. Bacteriocin Chemical Structure and Physicochemical Properties

Bacteriocins are heterogeneous group polypeptides with different morphological and biochemical properties. Figure 5 shows the tridimensional structures of cationic bacteriocins commonly used in the food industry. It can be noted that all of them have alpha helix domains, a key structural element in the activity of this class of biomolecules. In addition, Table 5 shows the physicochemical properties of these biomolecules where the average molecular weight is 4815.12 Da with a high positive charge, due to the presence of basic residues (arginine and lysine) that, in turn, results in its high isoelectric point. Bacteriocins used in the food industry present a hydrophobicity property that varies around 0.10, indicating that these substances have a slightly higher affinity for fatty environments. This property gives them good bioavailability characteristics and contributes to their successful application as a treatment against pathogens. The high hydrophobic moment, which has been reported to be an important descriptor for bioactivity, is due to the peptides’ well-defined alpha-helical regions (see Figure 3).

To summarize, the analysis of the physicochemical space of the bacteriocins, as shown in Figure 5 and Table 5, can serve as a guide to search for other molecules with similar properties or that could be more effective in modulating properties like their hydrophobicity. This would allow their potential modification (i.e., increasing it) to improve their bioavailability and timing. Hydrophobicity can also be potentiated to enhance its activity against pathogens.

## 7. Conclusions

The growing number of antibiotic-resistant bacteria has led to a decrease in the therapeutic effectiveness of some antibiotic treatments and a higher incidence of resistant bacterial infections. This phenomenon has detrimental effects on economic and social aspects of public health systems.

The development of ABR organisms is affected by practices in the healthcare and the agrifood sectors, as the presence of antibiotic residues in food can increase the prevalence of ABR.

Systematic testing procedures for antibiotic residues in the food supply chain need to be established. Testing provides essential information for better control and would help to prevent ABR gene transfer, the emergence of ABR pathogens, and other negative consequences for consumer health associated with ABR.

The risk of antibiotic residues and their consequences will continue to be a threat unless generalized regulation is established. The search for alternatives to traditional antibiotics, such as antimicrobial peptides and targeted therapies using bacteriocins, may help to reduce the advance of antibiotic resistance by providing safer, more environmentally friendly options for disease control.

## Figures and Tables

**Figure 1 antibiotics-12-00550-f001:**
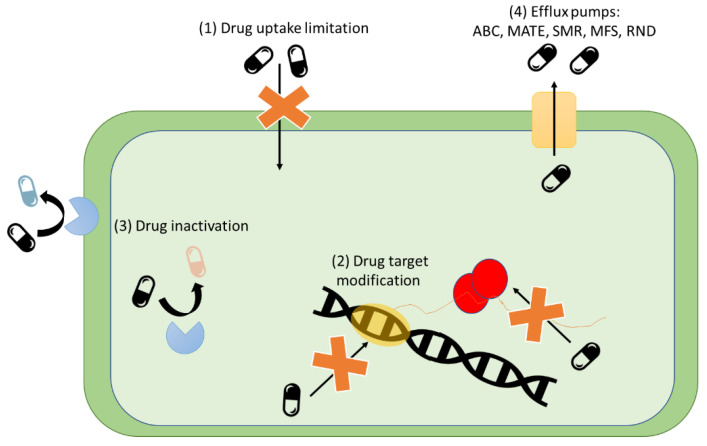
Graphical representation of antibiotic resistance mechanisms presents in bacteria: (**1**) Drug uptake limitation: decreased cell permeability to antimicrobial compounds reduces intake; (**2**) Drug target modification: cell modifies target molecules so that the antimicrobial no longer recognizes them; (**3**) Drug inactivation: cell enzymatic machinery modifies/inactivates antimicrobial molecules; (**4**) Efflux pumps: cell purges antimicrobial molecules through specific protein complexes (pumps). Adapted from [27].

**Figure 2 antibiotics-12-00550-f002:**
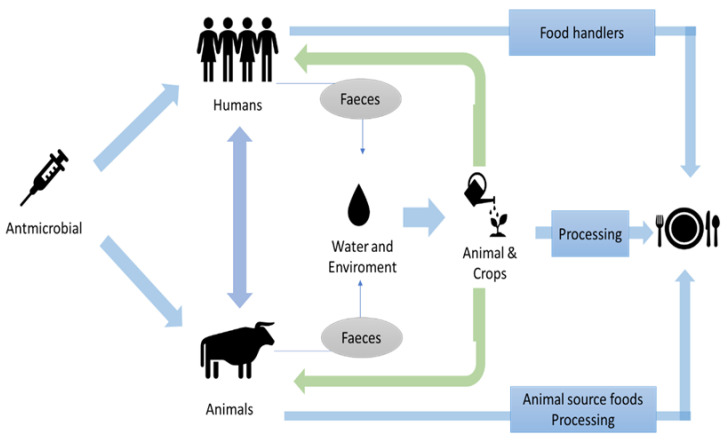
Potential pathways of ABR transmission throughout the food chain (adapted from [33]).

**Figure 3 antibiotics-12-00550-f003:**
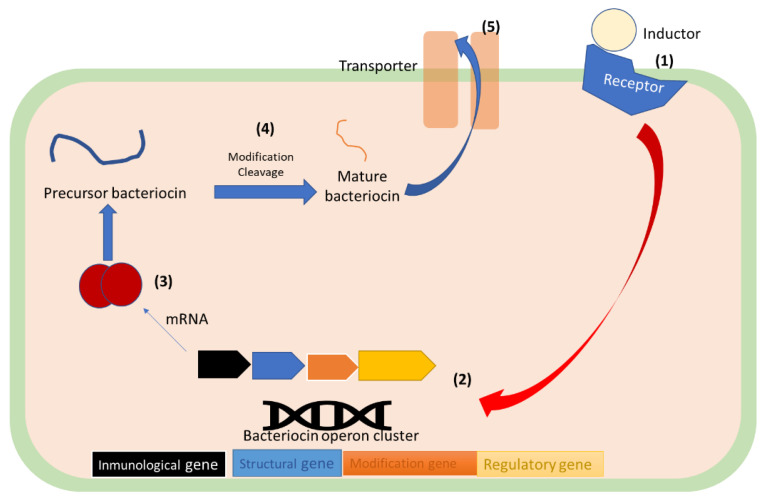
Graphical depiction of bacteriocin synthesis pathway: (**1**) and inductor induces de-expression of bacteriocin related genes; (**2**) Gene is transcribed to a mRNA; (**3**) mRNA is read by ribosomes, and the precursor bactericine is synthezised; (**4**) Precursor bacteriocin is modify and cleaved to form a mature bacteriocin; (**5**) Mature bacteriocin is excreted through a membrane transporter. (Original figure).

**Figure 4 antibiotics-12-00550-f004:**
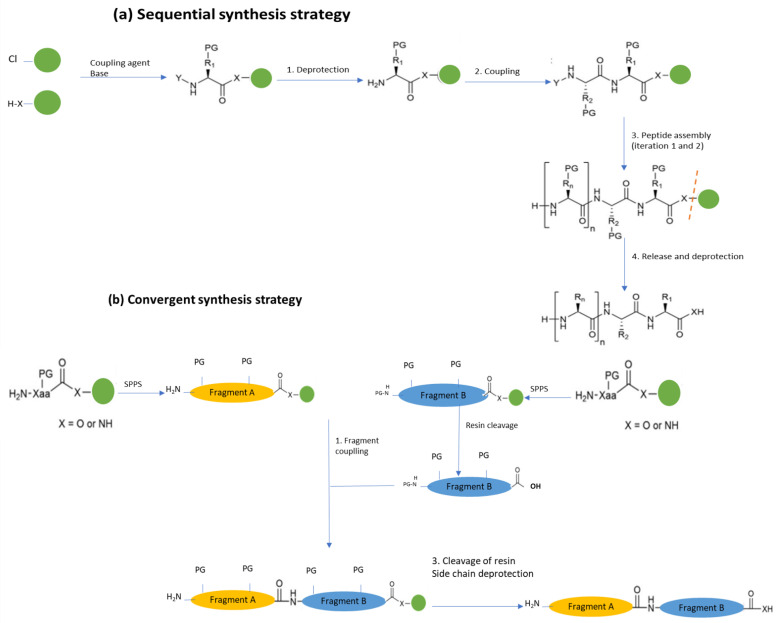
Graphical depiction of SPPS strategies: (**a**) lineal (sequential) synthesis based of iterative coupling and deprotection steps, and a final cleavage of solid phase; (**b**) Convergent synthesis of peptide fragments to generate a final polypeptide. Nomenclature is used as follows: Xaa = an undetermined amino acid, PG = protective groups; Solid phase = green circle. Adapted from [97].

**Figure 5 antibiotics-12-00550-f005:**
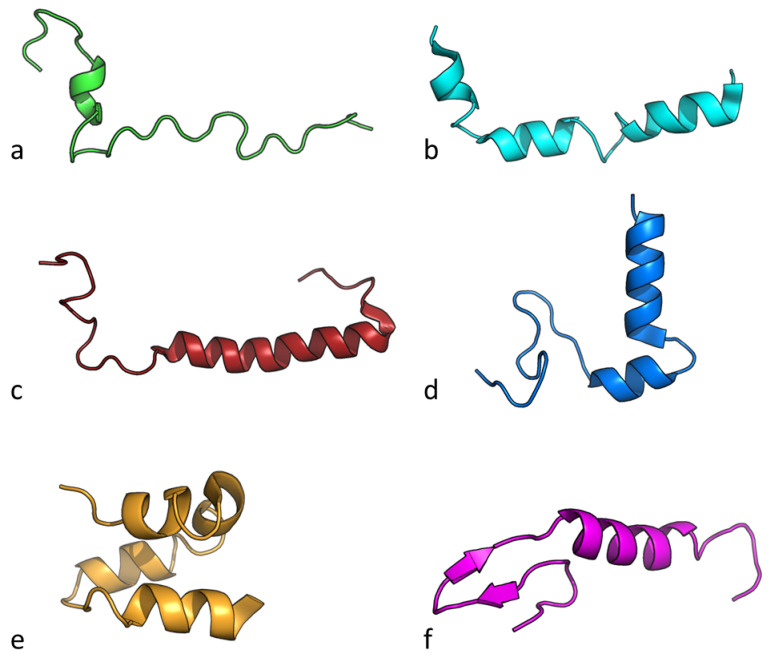
Structure of bacteriocins used in the food industry. Nisin Z (*Lactococcus lactis,* (**a**), lactococcin-G β *(Lactococcus lactis*, (**b**), Carnobacteriocin (*Carnobacterium maltaromaticum*, (**c**), Curvacin A (*Latilactobacillus curvatus*, (**d**), Enterocin 7A (*Enterococcus faecalis*, (**e**) and, Leucocin A (*Leuconostoc gelidum*, (**f**).

**Table 1 antibiotics-12-00550-t001:** Commonly used antibiotics and their associated resistance mechanisms.

Antimicrobial Group	Resistance Mechanism
Aminoglycosides Gentamicin Streptomycin Kanamycin	Enzyme modification Decreased permeability Target resistance (ribosome) Efflux pumps
β-Lactams Cephalothin Cefoxitin Ceftiofur Cefquinome	Reduced permeability Altered penicillin-binding proteins (PBPs) β-Lactamases, cephalosporinases Efflux pumps
Folate pathway inhibitors Sulfonamides	Decreased permeability Production of drug-insensitive enzymes
Macrolide-lincosamide-streptgramin B Erythromycin Lincomycin Virginiamycin	Enzyme modification Decreased permeability Decreased ribosomal binding
Phenicols Chloramphenicol Florfenicol	Enzyme modification Decreased permeability Decreased ribosomal binding Efflux pumps
Quinolones and fluoroquinolones Nalidixic acid Ciprofloxacin Enrofloxacin	Target resistance (DNA gyrase, topoisomerase IV) Efflux pumps Decreased permeability
Tetracyclines Chlortetracycline Tetracycline Doxycycline	Target resistance (ribosome) Drug detoxification Efflux pumps

**Table 2 antibiotics-12-00550-t002:** Antibiotic residues in animal-derived food products and associated health risks.

Antibiotic Residue	Concentration	Food Product	Associated Health Risks	Source
Oxytetracycline	2604.1 ± 703.7 μg/kg	Chicken muscle	Allergic hypersensitivity reactions or toxic effects (phototoxic skin reactions, chondrotoxic)	[44]
	3434.4 ± 604.4 μg/kg	Chicken liver	Carcinogenicity, cytotoxicity	
	51.8 ± 90.53 μg/kg	Beef	Carcinogenicity, cytotoxicity	[45]
Enrofloxacin	0.73–2.57 μg/kg	Chicken meat	Allergic hypersensitivity reactions or toxic effects, phototoxic skin reactions, chondrotoxic.	[47]
Chloramphenicol	1.34–13.9 μg/kg	Chicken	Bone marrow toxicity, optic neuropathy, brain abscess	
Penicillin	0.87–1.3 μg/kg	Veal	Allergy, affects starter cultures for fermented milk products	
Oxytetracycline	3.5–4.61 μg/kg	Chicken meat	Carcinogenicity, cytotoxicity in the bones of broiler chickens	
Quinolones	30.81 ± 0.45 μg/kg	Chicken meat	Allergic hypersensitivity reactions or toxic effects (phototoxic skin reactions, chondrotoxic)	[47]
	6.64 ± 1.11 μg/kg	Beef		
Amoxicillin	9.8–56.16 μg/mL	Milk	Carcinogenic, teratogenic, and mutagenic effects	[48]
	10.46–48.8 μg/g	Eggs		
Sulfonamides	16.28 μg/g	Raw milk	Carcinogenicity, allergic reaction	[49]
Quinolones	23.25 μg/g		Allergic hypersensitivity reactions or toxic effects (phototoxic skin reactions, chondrotoxic).	

**Table 3 antibiotics-12-00550-t003:** Prevalence of ABR strains in the food chain.

Microorganism	Sample Source	Antibiotic Resistance	Prevalence (%)	Source
*Escherichia coli*	Bovine milk sample	Azithromycin	53	[53]
		Chloramphenicol	15	
		Ceftriaxone	17	
		Penicillin	69	
		Gentamicin	6	
		Amoxicillin	55	
		Tetracycline	20	
		Cephalexin	64	
*Listeria monocytogenes*	Bovine milk sample	Azithromycin	12	
		Chloramphenicol	22	
		Ceftriaxone	17	
		Penicillin	46	
		Gentamicin	24	
		Amoxicillin	46	
		Tetracycline	23	
		Cephalexin	46	
*Salmonella* spp.	Bovine milk sample	Azithromycin	8	
		Chloramphenicol	6	
		Ceftriaxone	5	
		Penicillin	21	
		Amoxicillin	15	
		Tetracycline	5	
		Cephalexin	21	
*Staphylococcus aureus*	Bovine milk sample	Azithromycin	8	
		Chloramphenicol	6	
		Ceftriaxone	6	
		Penicillin	21	
		Gentamicin	3	
		Amoxicillin	25	
		Tetracycline	7	
		Cephalexin	25	
*E. coli*	Healthy farm workers	β-lactams	77.3	[54]
	Pigs		76.7	
	Poultry broilers		40	
*S. aureus*	Pigs	Methicillin	30	[55]
*Campylobacter jejuni*	Chicken	Ampicillin	5	[56]
		Tetracycline	31.7	
		Ciprofloxacin	23.3	
*C. coli*	Pork	Ampicillin	33.3	
		Erythromycin	73.3	
		Tetracycline	73.3	
		Chloramphenicol	6.7	
		Ciprofloxacin	46.7	

**Table 4 antibiotics-12-00550-t004:** List of some cationic bacteriocins and their uses in the food industry.

Bacteriocin	Source	Food Use	Reference
Nisin and Nisin Z	*Lactococcus lactis*	Prevents food spoilage caused by *Lactobacillus* spp., *L. monocytogenes*, *S. aureus,* and *Clostridium* spp.	[79]
lactococcin-G β	*Lactococcus lactis*	Activity against L. *monocytogenes* in yogurt, cheese, and sauerkraut	[84]
Leucocin A	*Leuconostoc gelidum*	Activity against *E. coli* and *L. monocytogenes* in meat and fish products.	[84]
Carnobacteriocin B2	*Carnobacterium maltaromaticum*	Activity against *L. monocytogenes* in dairy, meat, or fish food and feed products	[85]
Curvacin A	*Latilactobacillus curvatus*	Activity against *Listeria monocytogenes*	[86]
Enterocin 7A	*Enterococcus faecalis*	Activity against *L. monocytogenes* in meat and meat-based products	[87]

**Table 5 antibiotics-12-00550-t005:** Physicochemical properties of some cationic bacteriocins used in the food industry.

Name	Source Organism	Molecular Weight (Da)	Net Charge pH 7	Isoelectric Point	Hydrophobicity	Hydrophobic Moment
Nisin	*L. lactis*	3456.62	3	8.52	−0.29	0.48
lactococcin-G β	*L. lactis*	4107.19	4	10.42	0.25	0.71
Leucocin A	*L. gelidum*	3929.80	2	8.77	0.26	1.58
Carnobacteriocin B2	*C. maltaromaticum*	4966.40	4	9.96	0.00	1.60
Curvacin A	*L. curvatus*	4306.03	3	9.37	0.11	1.69
Enterocin 7A	*E. faecalis*	5172.91	6	10.68	0.20	2.12
Average		4815.12	4	10.00	0.10	1.81

## Data Availability

Not applicable.

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
