# Peer review of "Antibiotic Resistance and Food Safety: Perspectives on New Technologies and Molecules for Microbial Control in the Food Industry"

_antibiotics, 2023, doi:10.3390/antibiotics12030550_

Round 1
Reviewer 1 Report
I can confirm that I have reviewed a manuscript titled “Antibiotic resistance and food safety: Perspectives on new technologies and molecules for microbial control in the food industry”. Given that AMR is a serious global issues which requires prompt solution, the importance of this paper cannot be over-emphasized. However, I suggest that the authors must include two subheadings which describe the techniques or methods used to synthesize peptides. Once such information has been included, the paper may be accepted.
Author Response
In representation of the authors, I would like to thank you for your valuable critiques aimed at the improvement of this paper.
We have followed your suggestions and have added a Subheading with Bacteriocins as its main topic, as it is the main interest of the review. In this subheading, we have included general details about bacteriocin chemical characteristics, such as structure and classifications, and both production pathways through natural biosynthesis and chemical synthesis.
Section 6.1, onwards. Line 458 to 570
Reviewer 2 Report
The paper addresses the food safety impact of antimicrobial resistance and contains information that are rather unique and probably a substantial contribution to the literature, e.g., concentration of antibiotics in various samples. However, there are claims of about the negative effect of antibiotics that requires adequate referencing, especially at the first mention and a strong rational defence of the claims. The main ones are the carcinogenicity and mutagenic abilities of antibiotics. If these were known, clearly the drugs would not have been approved for use. Even if this were to be true, it would apply to a tiny fraction of currently used antibiotics and this also needs to be made clear in the manuscript.
The early part of the paper including the Abstract contains a lot of grammatical mistakes, missing words or inappropriately inserted characters. The other parts had many other not-so-serious but copious problems that included: inappropriate hyphenation, long sentences, punctuation errors, typos and inconsistent spelling.
The manuscript requires an improved standard of grammar and writing.
Author Response
In representation of the authors, I would like to thank you for your valuable critiques aimed at the improvement of this paper.
We have followed your suggestions and have rewritten our claims in a clearer way as well as included more literature to give background on each case. These issues with antibiotics are well known, in particular with some antibiotics such as chloramphenicol, sulfonamides, and fluoroquinolones which are known to be potentially carcinogenic, hence why the FDA regulates their use as best as possible in clinical use, however there still no information on how could the consumption of residues, if it was the case, would impact human health. Our intention was not to pinpoint antibiotics as disease-causing, but rather pinpoint the potential risks associated with exposure to residues on daily basis. We hope our revision satisfies your observations ( mainly lines 275 -288)
On the other hand, the language was checked thoroughly and read-proofed by a native speaker, as well.
Reviewer 3 Report
The paper centers on a vital global public health issue of antimicrobial resistance in the food chain and the resultant food safety and public health issues. However, issues raised and listed below need to be satisfactorily addressed before the manuscript could reach a publishable value.
Comments for authors
- Abstract: Lines 13-15: Rewrite the sentence for clarity and to convey the intended meaning.
- Lines 13-15: “… dependent on many environmental factors?
- Line 17-18: Revise the sentence for clarity and to convey the intended meaning.
- The abstract is scanty and poorly written.
- Line 29: Replace “appearance” with “development”
- Line 37-40: Include the use of critically important antimicrobial agents in animal agriculture and availability of essential antibiotics as over-the-counter drugs, especially in developing countries as additional factors contributing to ABR. You may wish to read and cite the suggested article below: Antimicrobial drug usage pattern in poultry farms in Nigeria: implications for food safety, public health and poultry disease management. Veterinaria Italiana, 57 (1): 5-12. doi: 10.12834/VetIt.2117.11956.1
- Line 47-49: Please provide a concise background on why AMPs do not develop ABR easily to guide readers’ discretion
- Line 56-58: The definition of antimicrobial/antibiotic resistance is very primitive and defective. No reference to the dose or concentration of the drug.
- Line 69: replace “engraved” with “intrinsic”
- Line 72-73: Start by mentioning the two main methods of acquiring of antimicrobial resistance – Vertical and horizontal gene transfers – before you explain the types and mechanism of horizontal gene transfer
- 2.1 Antibiotic resistance acquisition and transference mechanisms: It will the good to explain the roles of mobile gene elements – Plasmids, transposons and integrons in development and transmission of ABR in this section
- Figure 1: The figure legend is not descriptive enough. Figure legends should be able to stand alone and readers should be able have a full understanding the figure without reference to the body of the manuscript
- Figure 1: Did the author create the figure or what it adapted from another author? If the latter is the case, the authors should acknowledge or cite the source.
- Line 245-248: At least one citation is required. Table 2 is not an author. Consider and cite the recommended paper below for that statement. You can still:
Assessment of antimicrobial drug administration and antimicrobial residues in food animals in Enugu State, Nigeria, Tropical Animal Health and Production, 50:897-902. https://doi.org/10.1007/s11250-018-1515-9
- There no information on how the literatures cited, e.g Table 2 were search out. Provide a brief methodology on the literature search for the reproducibility of the work
- The authors should proofread the manuscript over and over again or ask a native English speaker for assistance. There are so many typos in the paper. Alternatively, the manuscript could benefit from the services of professional English language editors.
Author Response
In representation of the authors, I would like to thank you for your valuable critiques aimed at the improvement of this paper.
We have followed your suggestions and applied corrections accordingly:
1-6:
-The writing of the paragraph was revised, and new information was included as suggested.
-The critical antimicrobial agents used in animal and agriculture were not included in this section as it is part of section 4, which addresses potential routes of transmission and prevalence of ABR in the food chain, and the use of antibiotics in both animal and agricultural practices is directly described.
7: A description of why peptides are less likely to develop resistance was included as suggested. Line 42 - 47.
8. The definition of ABR was improved and updated. Line 63 -73.
9. Word was replaced.
10. Both Horizontal and vertical gene transfer were defined as an introduction to the horizontal routes. Line 89-93
11. A brief description of mobile elements were also included as suggested. Line 123 -143
12-13. The figure legend was fixed. Line 166 -170
14. Citation was fixed.
15. A section of methodology was included 52- 61
16. The document was edited and proof-read several times by the authors and an external peer, and a native English speaker editor to improve grammar and general written language.
Thank you very much for the valuable criticism.
Round 2
Reviewer 3 Report
The authors have improved the manuscript. However, in the Method section of this paper, it would have been best to use the 2020 PRISMA diagram to elucidate the search, eligibility and inclusion criteria for published papers included in this review.